# ENHANCING MULTIMODAL LLMS REASONING VIA PERCEPTION REWARD MODELING

## ABSTRACT

Reinforcement Learning with Verifiable Rewards (RLVR) has significantly improved the reasoning capabilities of large language models (LLMs). Recent research has also extended it to multimodal large language models (MLLMs) to enhance multimodal reasoning. However, through systematic error analysis, we find that while RLVR effectively reduces reasoning errors in MLLMs, it fails to address perceptual errors, which often lead to incorrect inference results. Limited visual perception is a major bottleneck in multimodal reasoning. To address this issue, we propose a novel visual perception-enhanced reward model that explicitly encourages accurate visual understanding as a prerequisite for reasoning. Specifically, our approach first incentivizes accurate visual perception prior to reasoning and then assigns a perception-based reward to reinforce correct understanding of the visual input. Extensive experiments on multiple multimodal reasoning benchmarks demonstrate that our approach effectively alleviates the perceptual bottleneck and promotes more reliable multimodal reasoning.

## 1 INTRODUCTION

Reinforcement Learning with Verifiable Rewards (RLVR) has recently advanced the reasoning ability of text-only LLMs, enabling strong performance on mathematical reasoning and code generation (Guo et al., 2025; Shao et al., 2024; Lambert et al., 2024). Standard RLVR deploys rule-based signals—format-consistency and answer-accuracy rewards—thereby reducing reliance on subjective preference models in RLHF (Christiano et al., 2017; Ouyang et al., 2022). Motivated by these successes, recent work extends RLVR to multimodal large language models (MLLMs) (Huang et al., 2025; Wang et al., 2025b; Liu et al., 2025a). Unlike text-only LLMs, however, MLLMs must first perceive visual content correctly before they can reason about it. When verification only checks the final answer or reasoning format, early perception errors can propagate and become irrecoverable.

To examine this mismatch, we audited 200 MLLM failures on We-Math (Qiao et al., 2024) before and after GRPO-style RLVR training (Shao et al., 2024); see Figure 1. Errors were labeled as perception and reasoning. Prior to RLVR, perception and reasoning each accounted for roughly half of failures. After training, reasoning errors decreased substantially, while perception errors remained essentially unchanged. Figure 2 illustrates a typical case: misreading a 35 degrees annotation leads to a wrong answer despite otherwise correct algebra. These findings identify perception as the dominant bottleneck after RLVR.

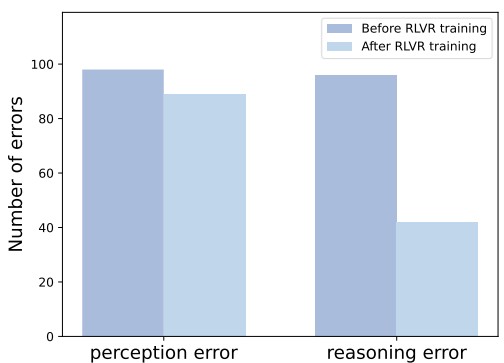

Figure 1: Comparison of the number of different error categories before and after RLVR training.

We therefore propose a perception-first training protocol that verifies perception before reasoning. The model first produces a perceptual summary of the image (an image caption) and then derives the answer conditioned on this summary. We assign an automated perception reward via pseudo–ground-truth (PGT) captions generated by a

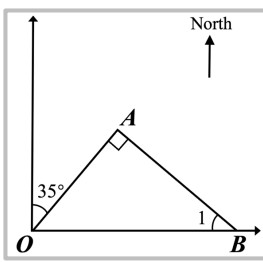

**Question:** As shown in the diagram, triangle ABO is a right triangle. What is the measure of ∠1 in degrees?
**Answer: 35**

**GRPO response**
<think>In a right triangle, the sum … 90 degrees. **We are given that one of the other angles is 35 degrees.** Let's denote the angle … 55 degrees; Therefore, **the measure of ∠1 is 55 degrees.**</think>    \boxed{55}

**Ours response**
<caption>The image shows a diagram ... Point A has a right angle. **The angle between the line extending north from point O and the line segment OA is 35 degrees.** The angle at point … labeled North.</caption>
<think>To solve for the measure of ∠1, we … ∠O= 90 - 35 = 55 degrees … ∠1 and the ∠O are complementary … 35 degrees. Thus, **the measure of ∠1 is 35 degrees.**</think>    \boxed{35}

Figure 2: An error case of perception error after RLVR training.

strong MLLM (e.g., Gemini (Gemini Team et al., 2023)) and an LLM judge that scores semantic consistency between model and PGT captions, yielding a fine-grained signal. Crucially, we couple perception and answer-accuracy rewards multiplicatively, operationalizing a logical AND: high reward is granted only when both perception and the final answer are correct.

Extensive experiments on MathVerse (Zhang et al., 2024), MathVista (Lu et al., 2023), MathVision (Wang et al., 2024), and We-Math (Qiao et al., 2024) show consistent gains over strong RLVR baselines. Our contributions are: (i) a large-scale error audit establishing perception as the post-RLVR bottleneck for MLLMs; (ii) a perception-first protocol with an automated perception reward; (iii) multiplicative coupling of perception and answer rewards; and (iv) state-of-the-art or strong results on multimodal math reasoning benchmarks.

## 2 PRELIMINARY

In this section, we introduce the foundational concepts and formalize Multi-modal RLVR, covering the task formulation, reward modeling, and related algorithms to achieve RLVR.

### 2.1 PROBLEM FORMULATION

This study focuses on the reasoning task of multimodal large language models (MLLMs). Formally, given a dataset $D = \{x_1, x_2, \ldots, x_n\}$, where each instance $x = (I, q, \hat{a})$ consists of an image $I$, a natural language question $q$, and the corresponding ground-truth answer $\hat{a}$. The model $\pi_\theta$ processes the input pair $(I, q)$ and generates an output $o$ that comprises a coherent chain of reasoning followed by a final prediction.

### 2.2 REWARD MODELING

Reinforcement Learning with Verifiable Rewards (RLVR) enhances the model's reasoning capabilities by providing a clear binary reward to the model using a verifiable, rule-based reward function. Rule-based reward functions typically include format rewards and accuracy rewards:

- **Format rewards**: This component evaluates whether the model's output adheres to the specified structural format. For instance, it verifies that the reasoning process is enclosed within `<think>` and `</think>` tags, and that the final answer is properly placed within `\boxed{}`.

- **Accuracy rewards**: This component assesses the correctness of the model's response. In mathematical reasoning tasks, for example, it typically checks whether the final answer matches the ground-truth value.

## 2.3 REINFORCE LEARNING ALGORITHM

Group Relative Policy Optimization (GRPO) (Shao et al., 2024) is a variant of Proximal Policy Optimization (PPO) (Schulman et al., 2017). Unlike standard PPO, GRPO estimates the advantage directly from the group reward scores, without the need for an additional critic model. This approach substantially reduces memory consumption and computational overhead, leading to a more efficient training process. Specifically, for a given multimodal input $(I, q, \hat{a})$, GRPO first samples a group of outputs $\{o_1, o_2, \cdots, o_G\}$ from the old policy model $\pi_{\theta_{old}}$. Then, the advantage of the $i$-th output is computed using a group of rewards $\{R_1, R_2, \cdots, R_G\}$ corresponding to a group of outputs within each group:

$$A_{i,t} = \frac{R_i - \text{mean}(R_1, R_2, \cdots, R_G)}{\text{std}(R_1, R_2, \cdots, R_G)}. \tag{1}$$

Similar to PPO, GRPO adopts a clipped objective and applies a KL penalty term to optimize the policy model $\pi_\theta$ by maximizing the following objective:

$$\mathcal{J}_{\text{GRPO}}(\theta) = \mathbb{E}_{(I,q,\hat{a}) \sim D, \{o_i\}_{i=1}^G \sim \pi_{\theta_{old}}(\cdot|I,q)} \left[ \frac{1}{G} \sum_{i=1}^{G} \frac{1}{|o_i|} \sum_{t=1}^{|o_i|} \left( \right. \right.$$
$$\left. \min\left( \frac{\pi_\theta(o_{i,t}|I,q,o_{i,<t})}{\pi_{\theta_{old}}(o_{i,t}|I,q,o_{i,<t})} A_{i,t}, \text{clip}\left( \frac{\pi_\theta(o_{i,t}|I,q,o_{i,<t})}{\pi_{\theta_{old}}(o_{i,t}|I,q,o_{i,<t})}, 1-\varepsilon, 1+\varepsilon \right) A_{i,t} \right) - \beta \mathbb{D}_{KL}(\pi_\theta||\pi_{\text{ref}}) \right) \right], \tag{2}$$

## 3 METHOD

In this section, we first present an error analysis of multimodal reasoning in Section 3.1. We then then detail our approach to perception reward modeling in Section 3.2, and finally introduce our policy optimization method in Section 3.3.

### 3.1 ERROR ANALYSIS OF MULTIMODAL REASONING

MLLMs must first correctly perceive visual content before performing logical reasoning. However, existing RLVR-based multimodal reasoning methods primarily emphasize the answer accuracy, while overlooking whether MLLMs correctly perceive the visual content. To investigate this, we analyzed 200 error cases of MLLMs on We-Math (Qiao et al., 2024) before and after training with the standard GRPO (Shao et al., 2024). We observed that before RLVR, perception and reasoning each accounted for about half of the failures. After training, reasoning errors decreased significantly, while perception errors remained largely unaddressed. These results highlight that limited visual perception has become a primary bottleneck after RLVR, hindering further development of multimodal reasoning.

### 3.2 PERCEPTION-ENHANCED REWARD MODELING

To address this limitation, we introduce a perception-first training protocol that explicitly validates perception accuracy before reasoning, thereby ensuring the correctness of visual understanding. Our method first incentivizes the model to generate accurate visual perceptions (e.g., image captions) prior to reasoning, and then use this to derive the final answer. We then compute a perception reward using pseudo–ground-truth (PGT) captions generated by a powerful MLLM and an LLM judge that scores semantic consistency between model and PGT captions, resulting in a fine-grained signal. Importantly, perception quality and answer accuracy are combined through multiplicative reward shaping—implementing a logical AND: only when both perception and reasoning are correct is a high reward assigned. The policy model is subsequently optimized using a reinforcement learning algorithm to reinforce robust and accurate visual perception.

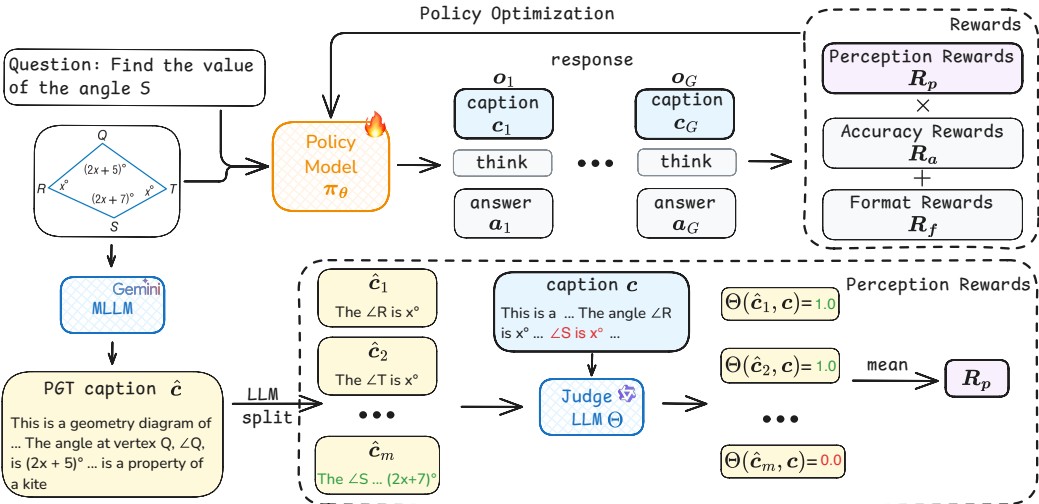

Figure 3: Overview of our method. The primary training pipeline is built upon RLVR Optimization Algorithms. In addition to conventional accuracy and format rewards, we introduce a novel perception reward. This reward is multiplicatively combined with the accuracy reward to enhance both the visual perception and reasoning capabilities of MLLMs.

### 3.2.1 MODEL RESPONSE FORMAT

To motivate the model to generate accurate visual perception before reasoning, we define the following input and output formats for MLLMs. Formally, given the input $x = (I, q, \hat{a}) \in D$, the model first describes the visual content of the image (i.e., generates a caption $c$) which is enclosed between `<caption>` and `</caption>` tags. The model then performs logical reasoning, enclosing the reasoning process within `<think>` and `</think>` tags, and finally places the generated answer $a$ within `\boxed{}` tag. We use a binary format reward $R_f \in \{-1, 1\}$ for evaluation to verify that output $o$ follows the caption-think-answer format.

### 3.2.2 PERCEPTION REWARD

To assign reliable visual reward scores to the generated captions, we employed a state-of-the-art MLLM model to generate high-quality captions for each image, which serve as pseudo-ground-truth (PGT) captions. These PGT captions provide a reliable and consistent standard for assigning visual perceptual reward signals. We then used an LLM model as a referee, leveraging its powerful comparative evaluation capabilities to score the semantic consistency between generated captions and PGT captions.

Specifically, as shown in Figure 3, for the input image $I \in D$, we prompt a strong MLLM (e.g., Gemini-2.5-Pro) to generate PGT caption $\hat{c}$. Since PGT captions typically contain multiple visual perceptual information, directly comparing generated captions with PGT captions will produce coarse-grained reward signals. To enable finer-grained perceptual reward assessment, we employ an LLM to decompose PGT captions into a set of simper captions $\{\hat{c}_1, \hat{c}_2, \cdots, \hat{c}_m\} \in \hat{c}$, where $\hat{c}_i$ contains less visual information. Then we use LLM $\Theta$ as a referee to judge the semantic consistency between the generated caption $c$ and a set of PGT captions $\{\hat{c}_1, \hat{c}_2, \cdots, \hat{c}_m\}$. $\Theta(\hat{c}_j, c) = 1.0$ indicates that the basic semantic information of PGT caption $\hat{c}_j$ is contained in the generated caption $c$, otherwise $\Theta(\hat{c}_j, c) = 0.0$. The perception reward $R_p$ is expressed as follows:

$$R_p = \frac{1}{m} \sum_{j=1}^{m} \Theta(\hat{c}_j, c), \text{ where } \Theta(\hat{c}_j, c) \in \{0, 1\} \tag{3}$$

### 3.2.3 PERCEPTION-ENHANCED REWARDS

After obtaining the visual perception reward, we do not simply add it to rule-based rewards. Instead, we combine it multiplicatively with the accuracy reward. Specifically, for the input $(I, q, \hat{a}) \in D$, we use a binary accuracy reward $R_a \in \{-1, 1\}$ for comparing whether the generated answer $a$ in the output $o$ is equal to ground-truth answer $\hat{a}$. We then multiply the visual perception reward $R_p$ with the accuracy reward $R_a$ and add the format reward $R_f$. The perception-enhanced reward function $R$ is then expressed as follows:

$$R = \begin{cases} \alpha \cdot R_p \cdot R_a + \beta \cdot R_f, & \texttt{is\_equivalent}(\hat{a}, a) \\ \alpha \cdot (1 - R_p) \cdot R_a + \beta \cdot R_f, & \text{otherwise} \end{cases} \tag{4}$$

where $\alpha$ and $\beta$ are hyperparameters that scale the contributions of the accuracy and format rewards. $\texttt{is\_equivalent}(\hat{a}, a) = 1$ means the generated answer $a$ and ground-truth answer $\hat{a}$ are equal. This reward function encourages correct visual perception more when the generated answer $a$ matches ground-truth answer $\hat{a}$, and punishes incorrect visual perception more harshly when $a$ is inconsistent with $\hat{a}$.

### 3.3 POLICY OPTIMIZATION WITH PERCEPTION-ENHANCED REWARDS

We follow the GRPO approach but introduce several modifications based on recent studies (Yu et al., 2025; Liu et al., 2025d). Specifically, we first replace the sample-level loss used in GRPO with a token-level loss to better learn reasoning patterns from high-quality long examples, suppress the impact of low-quality long samples, and improve training stability. We then apply a dynamic sampling strategy to ensure that there is at least one correct and incorrect sample in each batch, avoiding gradient vanishing, improving training stability, and reducing gradient variance. Furthermore, we remove the KL penalty, allowing the model to explore more freely and preventing it from being constrained by unnecessary constraints, thereby improving its ability to reason about complex problems. For each question $q$ and image $I$, we samples a group of outputs $\{o_1, o_2, \cdots, o_G\}$ from the old policy model $\pi_{\theta_{old}}$ and then optimizes the policy model $\pi_\theta$ by maximizing the following objective:

$$\mathcal{J}(\theta) = \mathbb{E}_{(I,q,\hat{a}) \sim D, \{o_i\}_{i=1}^G \sim \pi_{\theta_{old}}(\cdot|I,q)} \left[ \frac{1}{\sum_{i=1}^G |o_i|} \sum_{i=1}^G \sum_{t=1}^{|o_i|} \right.$$

$$\left. \min\left( \frac{\pi_\theta(o_{i,t}|I,q,o_{i,<t})}{\pi_{\theta_{old}}(o_{i,t}|I,q,o_{i,<t})} A_{i,t}, \text{clip}\left( \frac{\pi_\theta(o_{i,t}|I,q,o_{i,<t})}{\pi_{\theta_{old}}(o_{i,t}|I,q,o_{i,<t})}, 1 - \varepsilon_{low}, 1 + \varepsilon_{high} \right) A_{i,t} \right) \right] \tag{5}$$

$$s.t. \ 0 < |\{o_i \mid \texttt{is\_equivalent}(\hat{a}, o_i)\}| < G,$$

where $(1 - \varepsilon_{low}, 1 + \varepsilon_{high})$ is the clipping range of importance sampling ratio, and $\texttt{is\_equivalent()}$ indicates whether the prediction output $o_i$ by the model is equal to the ground truth $\hat{a}$. For the advantage $A_{i,t}$, we removed the standard deviation normalization term from the advantage to prevent gradient disparities across problems of varying difficulty during training; for example, easier problems exhibit smaller standard deviations, which amplifies their advantages and leads to higher gradients. This imbalance ultimately results in a problem-level difficulty bias. The advantage is therefore computed using a group of rewards $\{R_1, R_2, \cdots, R_G\}$ corresponding to the outputs within each group as follows:

$$A_{i,t} = R_i - \text{mean}(R_1, R_2, \cdots, R_G), \tag{6}$$

where $R_i$ is calculated using the reward function in Eq. 4.

## 4 EXPERIMENTS

In this section, we first introduce the details of our experimental settings in Section 4.1. We then conduct an ablation study in Section 4.2 to verify the effectiveness of our method, and compare with the main results in Section 4.3.

## 4.1 EXPERIMENTAL SETUP

**Training Dataset**. Our experiments use the Geometry3K (Lu et al., 2021) dataset, which contains 2.1K training samples. In addition, we prompt Gemini-2.5-Pro (Gemini Team et al., 2023) to generate image captions for these 2.1k training images as pseudo-ground-truth (PGT) captions, and then use Qwen2.5-32B (Qwen, 2024) to split these PGT captions.

**Evaluation**. We evaluate our method on multiple multimodal mathematical reasoning benchmarks, including Mathverse (Zhang et al., 2024), MathVista (Lu et al., 2023), MathVision (Wang et al., 2024), and We-Math (Qiao et al., 2024), and a visual perception benchmark, HallusionBench (Guan et al., 2024). During inference, we follow previous works (Liu et al., 2025a) to accelerate the inference process using vLLM (Kwon et al., 2023), apply greedy decoding with a temperature of 0.0, and use Gemini-2.0-Flash (Gemini Team et al., 2023) as the discriminative model to parse generated responses.

**Baseline**. To evaluate the effectiveness of our proposed method, we compare it with the following two baselines: (1) Base Model: the model directly performs chain-of-thought (CoT) reasoning without any training. (2) GRPO: the model is trained using standard GRPO under the same backbone architecture and training data as our method. In addition, we include comparisons with state-of-the-art approaches reported in the literature: (1) proprietary models: GPT-4o (Hurst et al., 2024), Claude3.5 (Anthropic, 2024) and Kimi1.5 (Kimi Team et al., 2025) (2) open source models: LLaVA-OneVision (Li et al., 2024), Qwen2.5-VL (Qwen, 2025), InternVL2.5 (Chen et al., 2024), Kimi-VL-16B (Kimi Team et al., 2025), URSA-8B (Luo et al., 2025) and Mulberry-7B (Yao et al., 2024) (3) Reasoning models: R1-VL (Zhang et al., 2025), Vision-R1 (Huang et al., 2025), R1-OneVision (Yang et al., 2025), OpenVLThinker (Deng et al., 2025), MM-Eureka (Meng et al., 2025), ThinkerLite-VL (Wang et al., 2025d), VLAA-Thinker (Chen et al., 2025a), NoisyRollout (Liu et al., 2025a) and Perception-R1-7B (Xiao et al., 2025).

**Implementation Details**. Following prior work (Wang et al., 2025d; Liu et al., 2025a), we adopt Qwen2.5-VL-7B-Instruct (Qwen, 2025) as the base model and train it using the EasyR1 (Yaowei Zheng, 2025) framework. During training, we use Qwen2.5-32B (Qwen, 2024) as the judge LLM. All experiments are conducted on 8 NVIDIA-H20-96G GPUs. We adopt the default settings from EasyR1, using a learning rate of $1e-6$, a global batch size of 128, a rollout batch size of 512, and a rollout temperature of 1.0. We sample 12 rollouts (i.e., $G = 12$ in Eq. 5) and train for a total of 25 epochs. The coefficients in Eq. 4 are set to $\alpha = 0.1$ and $\beta = 0.9$, respectively.

## 4.2 ABLATION STUDY

In this section, we analyzes the impact of different components of our method. We first evaluate the effectiveness of our proposed perception-enhanced reward model. Then, we evaluate the impact of policy optimization algorithm.

(1) **The effectiveness of perception-enhanced reward model**. To evaluate the effectiveness of our proposed perception-enhanced reward model, we compare our complete method with an ablated version that excludes this reward component, under the same experimental setup. As illustrated in Figure 4, removing the perception-enhanced rewards leads to a notable decline in performance on the MathVerse and MathVista reasoning benchmarks, as well as on the Hallusionbench perception benchmark. These results indicate that the proposed reward model effectively enhances the perceptual capabilities of multimodal large language models, and that improved perception further contributes to better reasoning performance.

We further evaluated our method on the MathVision and WeMath benchmarks. As shown in Figure 5, our approach achieved performance comparable to the version without the perception-enhanced rewards on the WeMath benchmark, but fell short on the MathVision benchmark. We hypothesize that the reason for this discrepancy is that MathVision contains challenging problems from math competitions that require stronger reasoning capabilities, which our current base model may struggle with due to its size.

(2) **Impact of policy optimization algorithm**. we follow Dr.GRPO (Liu et al., 2025d) removed the standard deviation normalization term $\text{std}(R_1, R_2, \cdots, R_G)$ from the advantage $A_{i,t}$ in Eq. 6 to prevent gradient differences between problems of different difficulty levels during training. To verify the impact of this strategy, we conducted experiments in Table 1. The experimental results

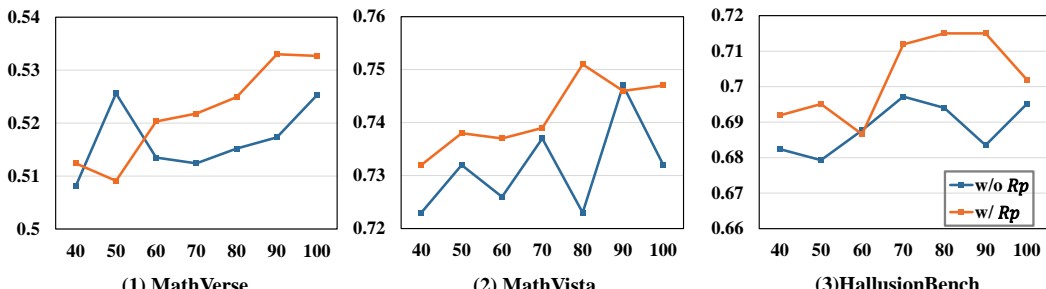

(1) MathVerse       (2) MathVista       (3)HallusionBench

Figure 4: Performance comparison of our complete method and our method without the perception-enhanced reward model on MathVerse, MathVista and HallusionBench benchmarks. The X-axis represents reinforcement learning training steps, the **blue** line represents the method without perception-enhanced reward model, and the **orange** line represents our complete method.

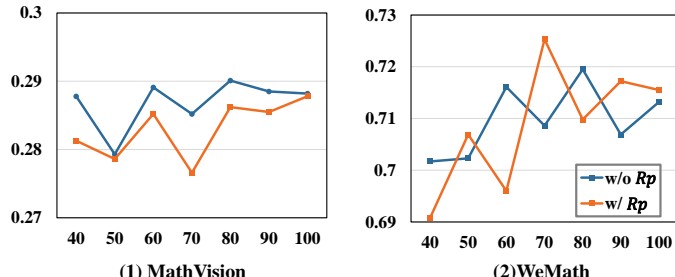

(1) MathVision       (2)WeMath

Figure 5: Performance comparison of our complete method and our method without the perception-enhanced reward model on MathVision and WeMath benchmarks. The X-axis represents reinforcement learning training steps, the **blue** line represents the method without perception-enhanced reward model, and the **orange** line represents our complete method.

show that removing the standard deviation normalization term significantly improves performance on all four benchmarks except the MathVison benchmark.

In addition to the difficulty bias at the question-level, we also studied whether the length bias at response-level has an impact on our work. We conducted experiments and the results are shown in Table 1. Experimental results show that removing the denominator normalization term from the response length $|o_i|$ in Eq. 5 does not does not improve performance. This may be because we added caption information to the response, which made the response very long and indirectly improved this bias.

Table 1: Ablation study of our method. All benchmarks report accuracy scores (%). The best value in each column is shown in **bold**.

| Model | MathVerse | MathVision | MathVista | WeMath | HallusionBench |
|---|---|---|---|---|---|
| **Ours** | **53.3** | 28.6 | **74.6** | **71.7** | **71.5** |
| Ours w/ $\text{std}(R_1, R_2, \cdots, R_G)$ in Eq. 6 | 51.9 | **28.8** | 73.5 | 71.0 | 69.1 |
| Ours w/o $\frac{1}{|o_i|}$ in Eq. 5 | 52.8 | 28.5 | 74.4 | 71.4 | 70.5 |

### 4.3 COMPARISON WITH STATE-OF-THE-ARTS

In this section, we presents a performance comparison between our method and existing state-of-the-art (SOTA) approaches across multiple multimodal reasoning benchmarks and one visual perception benchmark in Table 2.

Table 2: Performance comparison between our method with baselines on four benchmarks. All benchmarks report accuracy scores (%). Models marked with * are evaluated using our evaluation suite. The best value in each column is shown in **bold**, and the second-best value is underlined.

| Model | MathVerse | MathVision | MathVista | WeMath | HallusionBench |
|---|---|---|---|---|---|
| *Close-source models* | | | | | |
| GPT-4o | 50.8 | 30.4 | 63.8 | 69.0 | 71.4 |
| Claude-3.5-Sonnet | 26.5 | 38.0 | 67.7 | - | 71.6 |
| Kimi-k1.5 | - | 38.6 | 74.9 | - | - |
| *Open-source models* | | | | | |
| LLaVA-OneVision-7B | 26.2 | - | 63.2 | - | 48.4 |
| Mulberry-7B | - | - | 63.1 | - | - |
| InternVL2.5-8B | 39.5 | 19.7 | 64.4 | - | 67.3 |
| URSA-8B | 45.7 | 26.2 | 59.8 | - | - |
| Kimi-VL-16B | 44.9 | 21.4 | 68.7 | - | 66.2 |
| Qwen2.5-VL-72B-Instruct | - | 38.1 | 74.8 | - | 71.9 |
| InternVL2.5-78B | 51.7 | 32.2 | 72.3 | - | 72.9 |
| *Reasoning models* | | | | | |
| R1-VL-7B | 40.0 | 24.7 | 63.5 | - | - |
| Vision-R1-7B | 52.4 | - | 73.5 | - | - |
| R1-OneVision-7B | 46.1 | 22.5 | 63.9 | 62.1 | 65.6 |
| OpenVLThinker-7B | 48.0 | 25.0 | 71.5 | 67.8 | 70.8 |
| MM-Eureka-7B | 50.5 | 28.3 | 71.5 | 65.5 | 68.3 |
| ThinkLite-VL-7B | 50.2 | 27.6 | 72.7 | 69.2 | 71.0 |
| VLAA-Thinker-7B | 49.9 | 26.9 | 68.8 | 67.9 | 68.6 |
| NoisyRollout-7B* | 52.6 | **28.7** | 73.1 | 70.6 | 71.2 |
| Perception-R1-7B* | 50.1 | 28.4 | 73.3 | **72.6** | 68.6 |
| Base Model | 47.1 | 26.6 | 68.6 | 63.4 | 68.6 |
| GRPO | 51.1 | 27.9 | 71.0 | 68.2 | 68.9 |
| **Ours** | **53.3** | 28.6 | **74.6** | 71.7 | **71.5** |

Our method delivers strong performance across five visual reasoning and perception benchmarks. In particular on MathVerse, where our method achieves 53.3% accuracy, surpassing existing reasoning baselines and even surpassing InternVL2.5-78B and GPT-4o. On MathVista and HallussionBench, our method attains state-of-the-art accuracy scores of 74.6% and 71.5%, respectively. In addition, it also demonstrates competitive results on MathVision and WeMath.

We present a quantitative comparison of our method against two baselines: the Base Model and GRPO. As shown in Table 2, our approach achieves significant improvements across all benchmarks. Specifically, on MathVerse, it outperforms the Base Model and the vanilla GRPO by margins of 4.3% and 13.2%, respectively. On MathVista, the corresponding improvements are 8.7% and 5.1%. Furthermore, compared to the GRPO baseline, our method yields accuracy gains of 3.8% on HallusionBench, 2.5% on MathVision, and 5.1% on WeMath.

## 5 RELATE WORK

In this section, we first briefly summarize RLVR-based multimodal reasoning methods. Then, we introduce several reward models in reinforcement learning.

## 5.1 RLVR-BASED REASONING IN MLLMs

The success of DeepSeek-R1 (Guo et al., 2025) in enhancing model reasoning capabilities via Reinforcement Learning with Verifiable Rewards (RLVR) has spurred advances in multimodal reasoning. Several studies have since extended RLVR to various domains. For instance, VisualThinker-R1-Zero (Zhou et al., 2025) applied it to spatial reasoning, while MM-Eureka (Meng et al., 2025) adapted it to mathematical visual question answering. Other works have introduced algorithmic refinements: OThink-MR1 (Liu et al., 2025c) enhanced the GRPO algorithm by dynamically weighting the KL divergence term to balance exploration across training phases, and ThinkLite-VL (Wang et al., 2025d) incorporated a sample selection strategy based on Monte Carlo Tree Search (MCTS).

To enhance model reasoning performance and training efficiency, several methods adopt cold-start strategies. Vision-R1 (Huang et al., 2025), for example, uses DeepSeek-R1 to generate cold-start data for supervised fine-tuning (SFT) before applying a staged RL training process. Similarly, LMM-R1 (Peng et al., 2025) employs a two-stage training pipeline—first strengthening reasoning ability through RL on text-only data, then incorporating multimodal data to enhance multimodal reasoning. R1-Onevision (Yang et al., 2025) constructs a specialized reasoning dataset with structured image descriptions to encourage active use of visual information during reasoning. NoisyRollout (Liu et al., 2025a) improves exploration by mixing clean and noisy visual inputs.

Despite these advances, existing methods remain largely constrained by format and accuracy rewards, often overlooking the distinctive challenges inherent in multimodal reasoning. Our approach alleviates these challenges through the perception-enhanced reward model.

## 5.2 REWARD MODELING IN RL

Recognizing that the accuracy and format rewards of RLVR are ill-suited for the diverse challenges of multimodal reasoning, recent research has increasingly turned to task-specialized reward designs. For instance, Vision-RFT (Liu et al., 2025e) and Seg-Zero (Liu et al., 2025b) introduce specific visual rewards for fundamental perception tasks like classification, detection, and segmentation. For more complex structural and temporal reasoning, R1-SGG (Chen et al., 2025b) and VideoChat-R1 (Li et al., 2025b) employ modular rewards for scene graph generation and video understanding, respectively. Furthermore, Relation-R1 (Li et al., 2025a) designs relation rewards for visual relations, Video-R1 (Feng et al., 2025) introduces a temporal consistency reward, and VLAA-Thinker (Chen et al., 2025a) proposes a hybrid reward module to guide adaptive reasoning. We propose a visual perception-enhanced reward model that encourages accurate visual understanding before reasoning to alleviate the multimodal reasoning bottleneck caused by limited visual perception.

## 6 CONCLUSION

In this study, we conducted a large-scale error analysis and found that limited visual perception has become a major bottleneck after RLVR, hindering further progress in multimodal reasoning. To address this issue, we proposed a perception-enhanced reward model that incentivizes accurate visual perception before reasoning and then assigns perception-based rewards to reinforce the correct understanding of visual input. Our approach achieves significant performance improvements on multiple multimodal reasoning and perception benchmarks, promoting more reliable multimodal reasoning.

**Limitations and Future Work.** Despite demonstrating promising multimodal perception and reasoning abilities across multiple benchmarks, our method is subject to certain limitations. A primary concern is its dependence on state-of-the-art MLLMs to generate the pseudo-ground-truth captions for reward modeling, as these inherent visual errors of models could compromise the reward signal. Additionally, the reliance on a Judge LLM for reward scoring presents risks of reward hacking and potential biases. Since our approach focuses on solving visual perception problems, it can be extended to other multimodal tasks. Therefore, our future research directions will include extending the framework of visually enhanced rewards to encompass a wider spectrum of complex multimodal reasoning domains.

## ETHICS STATEMENT

This paper proposes a method to improve the reasoning of multimodal large language models. This research focuses on model optimization and therefore does not introduce new ethical risks beyond those inherent in large language models.

## REPRODUCIBILITY STATEMENT

In this article, we present the implementation details of our proposed method, including the base model, framework, dataset, and specific parameters employed in the experiments. The source code will be made publicly available upon acceptance to ensure reproducibility and promote further research.

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

## A  USAGE OF LLMS

The language in this article was polished using Large Language Models (LLMs) to correct grammatical errors and improve fluency. All LLMs-suggested edits were meticulously reviewed and approved by the authors to ensure they preserved the original meaning and academic integrity.

## B  DATASETS AND BENCHMARKS

### B.1  DATASETS

Geometry3K (Lu et al., 2021) is a dataset of 3K geometry problems with images and text, collected from various sources. We use 2101 samples for training in this work.

## B.2 BENCHMARKS

MathVerse (Zhang et al., 2024) is a comprehensive visual mathematics benchmark designed to evaluate the multimodal mathematical reasoning capabilities of Multimodal Large Language Models (MLLMs). We evaluate our method on testmini split of MathVerse.

MathVision (Wang et al., 2024) has collected 3040 high-quality math problems from real math competitions, covering 16 different math subjects and divided into 5 difficulty levels. We evaluate our method on test split of MathVision.

MathVista (Lu et al., 2023) is a benchmark for evaluating the visual mathematical reasoning capabilities of models. The dataset contains 6,141 samples covering 31 multimodal datasets. We evaluate our method on testmini split of MathVista.

WeMath (Qiao et al., 2024) is the first benchmark specifically designed to explore the problem-solving principles beyond end-to-end performance. We evaluate our method on testmini split of WeMath.

HallusionBench (Guan et al., 2024) is a comprehensive benchmark designed for the evaluation of image-context reasoning. We evaluate the perceptual ability of our method on HallusionBench.

## C ADDITIONAL EXPERIMENTS

### C.1 GENERALIZATION ANALYSIS OF OUR METHOD

To evaluate the generalization capability of our proposed method, we train the base model on the larger-scale ViRL39K (Wang et al., 2025a) dataset using our method. This dataset contains approximately 3.9k training samples, covering a wide range of categories such as non-geometric mathematics, geometric mathematics, diagrams, charts, documents, and spatial relationships, which significantly enhances the diversity and scale of the training data. We adopt the same configuration as in the main experiments, with the exception of using the open-source Qwen3-VL-32B-Instruct (Bai et al., 2025) model instead of Gemini-2.5-pro (Gemini Team et al., 2023) to generate pseudo-ground-truth (PGT) captions. We maintain the same settings as in the main experiments, but with a rollout batch size of 384 and train for 2 epochs.

To validate the enhanced perception capabilities of our approach, we further evaluate its performance on multiple perception benchmarks, including BLINK (Fu et al., 2024), MMbench (Liu et al., 2024), MME (Fu et al., 2025), HR-bench (Wang et al., 2025c), and VstarBench (Wu & Xie, 2023). We also incorporate comparisons with VL-Rethinker (Wang et al., 2025a), R1-VL (Zhang et al., 2025), Gemini-2.5-Pro (Gemini Team et al., 2023), and Qwen3-VL-32B-Instruct (Bai et al., 2025). The results are shown in Table 3, demonstrating the perception capability of our method.

### C.2 FURTHER ANALYSIS

In this section, we conduct further analysis of our method. Our pseudo-ground truth (PGT) captions are generated using Gemini-2.5-Pro. To investigate whether the performance gains originate from distilling knowledge from Gemini-2.5-Pro, we perform supervised fine-tuning (SFT) to evaluate the distillation effect on the base model. As shown in Table 4, the results confirm the effectiveness of our method.

## D CLARIFICATION ON METHODOLOGICAL CONTRIBUTION AND RELATED WORK

Our work shares similarities with Perception-R1 (Xiao et al., 2025) in that both utilize Gemini-2.5-Pro to process training data. However, while Perception-R1 employs Gemini-2.5-Pro to generate full responses to training questions and images—thereby leveraging both its visual and reasoning capabilities—our approach only inputs images and uses Gemini-2.5-Pro's visual understanding to generate pseudo-ground-truth (PGT) visual descriptions.

Table 3: Performance comparison between our method and baselines on ten benchmarks. All benchmarks report accuracy scores (%). Models marked with * are evaluated using our evaluation suite. The best value in each column is shown in **bold**, and the second-best value is underlined.

| Model | MathVerse | MathVision | MathVista | WeMath | HallusionBench | BLINK | MMBench1.1$_{(EN)}$ | MME$_{(sum)}$ | HRBench$_{(4K/8K)}$ | VstarBench |
|---|---|---|---|---|---|---|---|---|---|---|
| *Close-source models* | | | | | | | | | | |
| GPT-4o | 50.8 | 30.4 | 63.8 | 69.0 | 71.4 | - | 82.1 | - | - | - |
| Claude-3.5-Sonnet | 26.5 | 38.0 | 67.7 | - | 71.6 | - | 83.4 | - | - | - |
| Kimi-k1.5 | - | 38.6 | 74.9 | - | - | - | - | - | - | - |
| Gemini2.5-pro | 65.9 | 66.0 | 77.7 | - | 60.9 | 70.0 | 86.6 | - | - | - |
| *Open-source models* | | | | | | | | | | |
| LLaVA-OneVision-7B | 26.2 | - | 63.2 | - | 48.4 | 48.2 | 80.8 | - | - | - |
| Mulberry-7B | - | - | 63.1 | - | - | - | - | 2396.0 | - | - |
| InternVL2.5-8B | 39.5 | 19.7 | 64.4 | - | 67.3 | 54.8 | 83.2 | 2344.1 | - | - |
| URSA-8B | 45.7 | 26.2 | 59.8 | - | - | - | - | - | - | - |
| Kimi-VL-16B | 44.9 | 21.4 | 68.7 | - | 66.2 | 57.3 | 83.1 | - | - | - |
| Qwen2.5-VL-72B-Instruct | - | 38.1 | 74.8 | - | 71.9 | - | 88.0 | 2448.0 | - | - |
| InternVL2.5-78B | 51.7 | 32.2 | 72.3 | - | 72.9 | 63.8 | 88.5 | 2494.5 | - | - |
| Qwen3-VL-32B-Instruct | 76.8 | 63.4 | 83.5 | - | 63.8 | 67.3 | 88.9 | - | 82.9/77.8 | 85.3 |
| *Reasoning models* | | | | | | | | | | |
| R1-VL-7B | 52.2 | 28.2 | 74.3 | - | - | - | - | **2395.0** | - | - |
| Vision-R1-7B | 52.4 | - | 73.5 | - | - | - | - | - | - | - |
| R1-OneVision-7B | 46.1 | 22.5 | 63.9 | 62.1 | 65.6 | - | - | - | - | - |
| OpenVLThinker-7B | 48.0 | 25.0 | 71.5 | 67.8 | 70.8 | - | - | - | - | - |
| MM-Eureka-7B | 50.5 | 28.3 | 71.5 | 65.5 | 68.3 | - | - | - | - | - |
| ThinkLite-VL-7B | 50.2 | 27.6 | 72.7 | 69.2 | 71.0 | - | 81.4 | - | - | - |
| VLAA-Thinker-7B | 49.9 | 26.9 | 68.8 | 67.9 | 68.6 | - | - | - | - | - |
| NoisyRollout-7B* | 52.6 | 28.7 | 73.1 | 70.6 | 71.2 | 55.2 | 83.4 | 2376.2 | 58.4/48.1 | 67.0 |
| Perception-R1-7B* | 50.1 | 28.4 | 73.3 | **72.6** | 68.6 | 55.1 | 83.0 | 2355.5 | **60.7**/50.6 | 66.5 |
| VL-Rethinker-7B* | 52.6 | 30.6 | 74.2 | 68.1 | 70.1 | 55.0 | 82.4 | 2369.1 | 60.4/49.9 | 66.0 |
| Base Model | 47.1 | 26.6 | 68.6 | 63.4 | 68.6 | 55.0 | **83.7** | 2332.7 | 58.4/45.5 | 66.0 |
| GRPO | 51.1 | 27.9 | 71.0 | 68.2 | 68.9 | - | - | - | - | - |
| **Ours(Geo3K)** | **53.3** | 28.6 | 74.6 | 71.7 | **71.5** | 54.6 | 83.6 | 2363.0 | 58.6/48.3 | 64.9 |
| **Ours(ViRL39K)** | 52.1 | 29.7 | **75.2** | 71.6 | **71.5** | **56.6** | 83.2 | 2368.0 | 60.5/**51.1** | **69.1** |

Table 4: Performance comparison between our method and the distillation method (SFT) on multiple benchmarks. All benchmarks report accuracy scores (%). The best value in each column is shown in **bold**, and the second-best value is underlined.

| Model | MathVerse | MathVision | MathVista | WeMath | HallusionBench | BLINK | MMBench1.1$_{(EN)}$ | MME$_{(sum)}$ | HRBench$_{(4K/8K)}$ | VstarBench |
|---|---|---|---|---|---|---|---|---|---|---|
| Base Model | 47.1 | 26.6 | 68.6 | 63.4 | 68.6 | 55.0 | **83.7** | 2332.7 | 58.4/45.5 | 66.0 |
| SFT | 36.2 | 21.05 | 65.9 | 54.5 | 68.8 | 54.0 | 81.6 | 2356.4 | 59.4/49.1 | 63.4 |
| **Ours(Geo3K)** | **53.3** | 28.6 | 74.6 | **71.7** | 71.5 | 54.6 | 83.6 | 2363.0 | 58.6/48.3 | 64.9 |
| **Ours(ViRL39K)** | 52.1 | **29.7** | **75.2** | 71.6 | 71.5 | **56.6** | 83.2 | **2368.0** | **60.5/51.1** | **69.1** |

Moreover, our method first guides the model to follow a "description → reasoning → answer" process, encouraging it to produce accurate visual perception. Then, we apply a multiplicative coupling mechanism that integrates visual perception rewards with answer correctness rewards. This mechanism assigns higher rewards to correct answers that are based on accurate perception, thereby more effectively promoting learning.

# E    TEMPLATES

In this section, we provide the prompt for training as shown in Figure 6, and the prompt as shown in Figure 7 for LLM Θ in Eq. 3 to judge the semantic consistency between the generated captions and the pseudo-ground-truth captions. And we also provide the prompt for generate pseudo-ground-truth captions as shown in Figure 8.

# F    CASE STUDY

In this section, we demonstrate the improved perception and reasoning capabilities of our approach compared to GRPO through two case studies, as shown in Figures 9 and Figures 10, respectively. From the case study in Figure 9, we can see that our method keenly captures a small brown object hidden in the background during the caption output stage, which is not observed in GRPO.

---

**Prompt for Training**

You FIRST generate a detailed caption based on the image, then use the image caption as a guide to think about the reasoning process as an internal monologue, and finally provide the final answer. The image caption MUST BE enclosed within <caption> </caption> tags. The reasoning process MUST BE enclosed within <think> </think> tags. The final answer MUST BE put in \boxed{}.

---

Figure 6: The prompt for training in our work.

---

**Prompt for LLM Θ to judge the semantic consistency**

For each Visual Information item, determine whether Qwen Caption omits or misinterprets key information (e.g., object, relationship, location, label, number, etc.).
Scoring rules:
   - consistent or only minor differences: 1.0
   - omissions or errors: 0.0
The Consistency Score MUST BE enclosed within <score1> </score1> tags, <score2> </score2> tags, etc.
The final output format should be:
Consistency Score: <score1> </score1>, <score2> </score2>, etc
For example:
Consistency Score: <score1>1.0</score1>, <score2>0.0</score2>

---

Figure 7: The prompt for LLM Θ to judge the semantic consistency between the generated captions and the pseudo-ground-truth captions.

---

**Prompt for generate pseudo-ground-truth captions**

Generate a detailed caption based on the image.

**Prompt for split pseudo-ground-truth captions**

Extract the Visual Information from the Gemini caption. Focus on visual elements such as objects, relationships, positions, labels, numbers, or structures inferred from the diagram. Provide only the essential details to understand the diagram, not the results of reasoning. Make sure the visual key information is written in English.

Visual Key Information MUST BE enclosed within <info1> </info1> tags, <info2> </info2> tags, etc.

---

Figure 8: The prompt for generate pseudo-ground-truth captions.

**Question:** Subtract all red matte cubes. Subtract all small green metal objects. How many objects are left?

**Answer**: 7

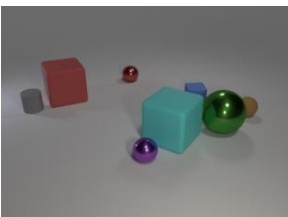

**GRPO response:**

<think>Let's break down the objects in the image: 1.There is one red matte cube. 2.There are no small green metal objects in the image. 3.There are other objects: a gray cylinder, a red shiny sphere, a blue matte cube, a cyan matte cube, a green shiny sphere, and a purple shiny sphere. Now, let's subtract the red matte cube:- After removing the red matte cube, we are left with 6 objects. Since there are no small green metal objects, no additional objects are removed. So, the total number of objects left is **6**.</think>

\boxed{**6**}

**Ours response:**

<caption>The image features a collection of 3D geometric shapes with various colors and materials. There are cubes, spheres, and a cylinder, each with distinct colors like red, cyan, blue, green, purple, and brown. The objects vary in size and reflect different materials, some appearing matte while others are shiny. The arrangement is spread out on a neutral background, highlighting the shapes and their properties.</caption>

<think>To solve the problem, let's identify and subtract the specified objects step by step: 1.Identify and subtract all red matte cubes: There is one red matte cube in the image. Subtracting this leaves the remaining objects. 2.Identify and subtract all small green metal objects: There are no small green metal objects in the image. Thus, no objects are subtracted in this step. After these steps, the remaining objects are: -A gray cylinder -A red matte cube (already subtracted) -A small red shiny sphere -A cyan matte cube -A small blue matte cube -A green shiny sphere -A small purple shiny sphere -A small brown matte sphere. Counting the remaining objects: -Gray cylinder: 1 -Small red shiny sphere: 1 -Cyan matte cube: 1 -Small blue matte cube: 1 -Green shiny sphere: 1 -Small purple shiny sphere: 1 -Small brown matte sphere: 1 Total remaining objects: **7**</think>

\boxed{**7**}

Figure 9: Case study demonstrate that our approach has better perception capabilities.

**Question:** As shown in the diagram, the circumference of the base of a cylindrical drum is 5cm. When the cylindrical drum rolls one complete revolution on the ground, the shape it traces is ( )?
**Options:**
  -A. Circle
  -B. Square
  -**C. Rectangle**
  -D. No correct answer
**Answer**: C

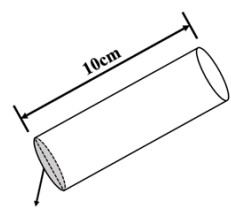

**GRPO response:**

**\<think\>** When a cylindrical drum rolls one complete revolution on the ground, the shape it traces is a circle. This is because the circumference of the base of the cylinder is the distance the drum travels in one complete revolution along the ground. The shape traced is a circle with a radius equal to the radius of the base of the cylinder. The circumference C of the base of the cylinder is given as 5 cm. The shape traced by the drum when it rolls one complete revolution is a circle with a circumference of 5 cm. Therefore, the correct answer is A. **Circle.\</think\>**
**\boxed{A}**

**Ours response:**

**\<caption\>** The image shows a cylindrical drum with labeled dimensions. It includes a length of 10 cm and a circumference C of 5 cm marked on the base. The task is to determine the shape traced by the drum when it rolls one complete revolution on the ground.**\</caption\>**
**\<think\>** To solve this, we need to understand the path traced by the cylindrical drum as it rolls. When a cylindrical object rolls without slipping, the path it traces is determined by the circumference of its base and its length. 1. The circumference of the base of the cylinder is given as 5 cm. This means that for every complete revolution, the drum travels a distance equal to its circumference. 2. The length of the cylinder is 10 cm, which represents the distance it travels along the ground in one complete revolution. When the cylinder rolls one complete revolution, it traces a path that forms a rectangle. The length of this rectangle is the length of the cylinder (10 cm), and the width is the circumference of the base (5 cm). Thus, the shape traced is a **rectangle.\</think\>**
**\boxed{C}**

Figure 10: Case study demonstrate that our approach has better reasoning capabilities.

