# OpenReview forum: "Enhancing Multimodal LLMs Reasoning via Perception Reward Modeling"
_ICLR.cc/2026/Conference — Submitted to ICLR 2026_

### Official Review · Reviewer_ay8z · 2025-10-25

**Soundness:** 2
**Presentation:** 3
**Contribution:** 2
**Rating:** 4
**Confidence:** 3

**Summary:**

This paper identifies that while Reinforcement Learning with Verifiable Rewards (RLVR) improves reasoning in multimodal models, it fails to fix errors in visual perception. To solve this, the authors propose a new reward model that explicitly incentivizes accurate visual understanding before reasoning. Their experiments show this method successfully alleviates the perception bottleneck, leading to more reliable multimodal reasoning.

**Strengths:**

- It provides an error analysis that pinpoints the limitations of existing RLVR methods—reasoning errors decreased significantly, while perception errors remained largely unaddressed.

- The introduced visual perception-enhanced reward model is a straightforward but effective solution, designed to incentivize accurate visual understanding before reasoning, which addresses the identified core problem.

- The approach is validated through experiments on multiple benchmarks, providing concrete evidence that the method effectively alleviates the perceptual bottleneck and improves the reliability of multimodal reasoning.

**Weaknesses:**

The pseudo-groundtruth captions were obtained using Gemini 2.5 Pro, but Gemini 2.5 Pro's performance on these benchmarks is not listed in Table 2. How does Gemini perform on these datasets? Would Gemini also encounter the perception errors during chain-of-thought reasoning?

Does the quality of pseudo-groundtruth captions affect RL training? How can the quality of pseudo-ground truth be validated? Is the reward signal from LLM-as-judge accurate?

**Questions:**

See the Weaknesses

---

> ### Author Response · Authors · 2025-11-23
>
> We wish to express our sincere appreciation for your insightful suggestions, which have guided significant improvements to our work. Detailed responses to each point are provided below, and we would appreciate your guidance on whether the revisions are satisfactory.
>
> **W1**: The pseudo-groundtruth captions were obtained using Gemini 2.5 Pro, but Gemini 2.5 Pro's performance on these benchmarks is not listed in Table 2. How does Gemini perform on these datasets? Would Gemini also encounter the perception errors during chain-of-thought reasoning?
>
> **A**: We have included the evaluation results based on Gemini-2.5-Pro. Our manual inspection of several generated captions revealed instances of perceptual errors. However, due to the labor-intensive nature of manually identifying and correcting these errors, we retained all original outputs without filtration or modification for the present paper. Addressing this limitation will be a focus of future work.
>
> | **Model** | **MathVerse** | **MathVision** | **MathVista** | **WeMath** | **HallusionBench** | **BLINK** | **MMBench1.1(EN)** | **MME(sum)** | **HRBench(4K/8K)** | **VstarBench** |
> |:---:|:---:|:---:|:---:|:---:|:---:|:---:|:---:|:---:|:---:|:---:|
> | **Close-source models** | | | | | | | | | | |
> | GPT-4o | 50.8 | 30.4 | 63.8 | 69.0 | 71.4 | - | 82.1 | - | - | - |
> | Claude-3.5-Sonnet | 26.5 | 38.0 | 67.7 | - | 71.6 | - | 83.4 | - | - | - |
> | Kimi-k1.5 | - | 38.6 | 74.9 | - | - | - | - | - | - | - |
> | Gemini2.5-pro | 65.9 | 66.0 | 77.7 | - | 60.9 | 70.0 | 86.6 | - | - | - |
> | **Open-source models** | | | | | | | | | | |
> | LLaVA-OneVision-7B | 26.2 | - | 63.2 | - | 48.4 | 48.2 | 80.8 | - | - | - |
> | Mulberry-7B | - | - | 63.1 | - | - | - | - | 2396.0 | - | - |
> | InternVL2.5-8B | 39.5 | 19.7 | 64.4 | - | 67.3 | 54.8 | 83.2 | 2344.1 | - | - |
> | URSA-8B | 45.7 | 26.2 | 59.8 | - | - | - | - | - | - | - |
> | Kimi-VL-16B | 44.9 | 21.4 | 68.7 | - | 66.2 | 57.3 | 83.1 | - | - | - |
> | Qwen2.5-VL-72B-Instruct | - | 38.1 | 74.8 | - | 71.9 | - | 88.0 | 2448.0 | - | - |
> | InternVL2.5-78B | 51.7 | 32.2 | 72.3 | - | 72.9 | 63.8 | 88.5 | 2494.5 | - | - |
> | Qwen3-VL-32B-Instruct | 76.8 | 63.4 | 83.5 | - | 63.8 | 67.3 | 88.9 | - | 82.9/77.8 | 85.3 |
> | **Reasoning models** | | | | | | | | | | |
> | R1-VL-7B | 52.2 | 28.2 | 74.3 | - | - | - | - | **2395.0** | - | - |
> | Vision-R1-7B | 52.4 | - | 73.5 | - | - | - | - | - | - | - |
> | R1-OneVision-7B | 46.1 | 22.5 | 63.9 | 62.1 | 65.6 | - | - | - | - | - |
> | OpenVLThinker-7B | 48.0 | 25.0 | 71.5 | 67.8 | 70.8 | - | - | - | - | - |
> | MM-Eureka-7B | 50.5 | 28.3 | 71.5 | 65.5 | 68.3 | - | - | - | - | - |
> | ThinkLite-VL-7B | 50.2 | 27.6 | 72.7 | 69.2 | 71.0 | - | 81.4 | - | - | - |
> | VLAA-Thinker-7B | 49.9 | 26.9 | 68.8 | 67.9 | 68.6 | - | - | - | - | - |
> | NoisyRollout-7B* | 52.6 | 28.7 | 73.1 | 70.6 | 71.2 | 55.2 | 83.4 | 2376.2 | 58.4/48.1 | 67.0 |
> | Perception-R1-7B* | 50.1 | 28.4 | 73.3 | **72.6** | 68.6 | 55.1 | 83.0 | 2355.5 | **60.7**/50.6 | 66.5 |
> | VL-Rethinker-7B* | 52.6 | **30.6** | 74.2 | 68.1 | 70.1 | 55.0 | 82.4 | 2369.1 | 60.4/49.9 | 66.0 |
> | Base Model | 47.1 | 26.6 | 68.6 | 63.4 | 68.6 | 55.0 | **83.7** | 2332.7 | 58.4/45.5 | 66.0 |
> | GRPO | 51.1 | 27.9 | 71.0 | 68.2 | 68.9 | - | - | - | - | - |
> | **Ours(Geo3K)** | **53.3** | 28.6 | 74.6 | 71.7 | **71.5** | 54.6 | 83.6 | 2363.0 | 58.6/48.3 | 64.9 |
> | **Ours(ViRL39K)** | 52.1 | 29.7 | **75.2** | 71.6 | **71.5** | **56.6** | 83.2 | 2368.0 | 60.5/**51.1** | **69.1** |
>
> **W2**: Does the quality of pseudo-groundtruth captions affect RL training? How can the quality of pseudo-ground truth be validated? Is the reward signal from LLM-as-judge accurate?
>
> **A**: The quality of pseudo-ground-truth captions indeed has impact on RL training performance. However, we did not assess the quality of the pseudo-ground-truth labels due to the significant time investment required for manual verification. Additionally, as mentioned in the limitations section, using LLMs as reward models can lead to reward hacking, and we will improve this in our future work.

---

### Official Review · Reviewer_Yw6V · 2025-11-01

**Soundness:** 4
**Presentation:** 2
**Contribution:** 3
**Rating:** 4
**Confidence:** 3

**Summary:**

This paper targets a concrete gap in current RLVR-style training for multimodal LLMs: after RLVR, reasoning errors drop but perception errors persist and become the new bottleneck. The authors first do an error audit on We-Math to show this shift, then propose a perception-first RL pipeline: the model must (1) produce a structured visual caption \<caption\>...\</caption\> before (2) reasoning \<think\>...\</think\> and (3) answering \boxed{}; a perception reward is computed by comparing the model’s caption to pseudo–ground-truth (PGT) captions generated by a strong MLLM and decomposed by an LLM judge; finally, the perception reward is multiplicatively coupled with the usual RLVR accuracy reward so that only “see correctly + answer correctly” gets high reward. On MathVerse, MathVista, MathVision, We-Math and HallusionBench, the method outperforms the base model and GRPO under the same backbone and is competitive with recent reasoning-style MLLMs.

**Strengths:**

- Empirical gains on several public benchmarks: Improvements are consistent on MathVerse, MathVista and HallusionBench, and the method stays competitive on We-Math and MathVision compared to strong open and closed models; this shows the idea is not tied to a single dataset.
- The paper is well-written and clearly structured. The narrative, from problem identification to solution and evaluation, is logical and easy to follow.

**Weaknesses:**

- Conceptual overlap with prior work [1]– The core idea of introducing a visual perception reward for improving multimodal reasoning has already been presented in Xiao et al., 2025, “Advancing Multimodal Reasoning Capabilities of MLLMs via Visual Perception Reward”.
Both works share the same motivation (perception as the bottleneck after RLVR) and similar implementation principles (PGT-based visual supervision + reward integration). The authors should clearly articulate what is novel here—for example, whether the contribution lies in (a) the multiplicative reward coupling, (b) the structured “caption→reason→answer” output enforcement, or (c) deeper empirical diagnosis. Without such clarification, the originality claim remains ambiguous.
- Training focuses on geometry-style math images. It is unclear whether the perception reward generalizes to other multimodal settings such as charts, UI, or natural images.
- The paper situates itself within the RLVR/R1 line but does not thoroughly compare with contemporaneous works such as VL-Rethinker or R1-VL that also incorporate self-reflective or vision-aligned objectives.

[1]Perception-R1: Advancing Multimodal Reasoning Capabilities of MLLMs via Visual Perception Reward.

**Questions:**

To strengthen the paper, the authors should (i) clarify the novelty boundary relative to that prior work, (ii) demonstrate robustness under open-source captioners and cheaper reward pipelines, and (iii) include a brief cross-domain experiment to confirm generality. Addressing these points would considerably improve both the clarity and contribution of the paper.
- How does this work substantively differ from prior work[1]. which also employs a visual perception reward? Is the main novelty the reward coupling mechanism or the structured generation format?
- Could the proposed perception reward extend to chart QA or document VQA tasks, where the caption length or structure differs significantly from diagram-based inputs?



[1]Perception-R1: Advancing Multimodal Reasoning Capabilities of MLLMs via Visual Perception Reward.

**Details Of Ethics Concerns:**

The paper presents a well-motivated and empirically solid reinforcement learning framework that explicitly enhances visual perception in multimodal reasoning. The approach is sound and yields consistent improvements; however, its conceptual and methodological proximity to prior work[1] significantly weakens the originality claim.

[1]Perception-R1: Advancing Multimodal Reasoning Capabilities of MLLMs via Visual Perception Reward.

---

> ### Author Response · Authors · 2025-11-23
>
> We are grateful for your constructive feedback, which has helped us enhance the manuscript. Our point-by-point revisions are listed below, and we hope they fully address your concerns.
>
> **W1**: Conceptual overlap with prior work [1]– The core idea of introducing a visual perception reward for improving multimodal reasoning has already been presented in “Advancing Multimodal Reasoning Capabilities of MLLMs via Visual Perception Reward”. Both works share the same motivation (perception as the bottleneck after RLVR) and similar implementation principles (PGT-based visual supervision + reward integration). The authors should clearly articulate what is novel here—for example, whether the contribution lies in (a) the multiplicative reward coupling, (b) the structured “caption→reason→answer” output enforcement, or (c) deeper empirical diagnosis. Without such clarification, the originality claim remains ambiguous.
>
> **Q1**: How does this work substantively differ from prior work[1]. which also employs a visual perception reward? Is the main novelty the reward coupling mechanism or the structured generation format?
> [1] https://arxiv.org/abs/2506.07218
>
> **A**: Our work shares certain similarities with [1]; however, the key differences lie in the following aspects:
>
> 1. While [1] leverages both the visual and reasoning capabilities of Gemini-2.5-Pro, we utilize only its visual ability to generate pseudo ground-truth visual descriptions of images.
> 2. Our approach first encourages the model to produce accurate visual perception following a "description → reasoning → answer" process. It then applies a multiplicative coupling mechanism that combines visual perception rewards with correctness rewards, providing high rewards for correct answers based on correct perception to enhance learning more effectively.
>
> **W2**: Training focuses on geometry-style math images. It is unclear whether the perception reward generalizes to other multimodal settings such as charts, UI, or natural images.
>
> **Q2**: Could the proposed perception reward extend to chart QA or document VQA tasks, where the caption length or structure differs significantly from diagram-based inputs?
>
> **A**: Our model was trained on the ViRL39K dataset, which contains diverse data types such as non-geometric mathematics, geometric mathematics, diagrams, charts, documents, spatial relationships, and others. Results across multiple benchmarks demonstrate the strong generalization capability of our approach.
>
> | **Model** | **MathVerse** | **MathVision** | **MathVista** | **WeMath** | **HallusionBench** | **BLINK** | **MMBench1.1(EN)** | **MME(sum)** | **HRBench(4K/8K)** | **VstarBench** |
> |:---:|:---:|:---:|:---:|:---:|:---:|:---:|:---:|:---:|:---:|:---:|
> | **Close-source models** | | | | | | | | | | |
> | GPT-4o | 50.8 | 30.4 | 63.8 | 69.0 | 71.4 | - | 82.1 | - | - | - |
> | Claude-3.5-Sonnet | 26.5 | 38.0 | 67.7 | - | 71.6 | - | 83.4 | - | - | - |
> | Kimi-k1.5 | - | 38.6 | 74.9 | - | - | - | - | - | - | - |
> | Gemini2.5-pro | 65.9 | 66.0 | 77.7 | - | 60.9 | 70.0 | 86.6 | - | - | - |
> | **Open-source models** | | | | | | | | | | |
> | LLaVA-OneVision-7B | 26.2 | - | 63.2 | - | 48.4 | 48.2 | 80.8 | - | - | - |
> | Mulberry-7B | - | - | 63.1 | - | - | - | - | 2396.0 | - | - |
> | InternVL2.5-8B | 39.5 | 19.7 | 64.4 | - | 67.3 | 54.8 | 83.2 | 2344.1 | - | - |
> | URSA-8B | 45.7 | 26.2 | 59.8 | - | - | - | - | - | - | - |
> | Kimi-VL-16B | 44.9 | 21.4 | 68.7 | - | 66.2 | 57.3 | 83.1 | - | - | - |
> | Qwen2.5-VL-72B-Instruct | - | 38.1 | 74.8 | - | 71.9 | - | 88.0 | 2448.0 | - | - |
> | InternVL2.5-78B | 51.7 | 32.2 | 72.3 | - | 72.9 | 63.8 | 88.5 | 2494.5 | - | - |
> | Qwen3-VL-32B-Instruct | 76.8 | 63.4 | 83.5 | - | 63.8 | 67.3 | 88.9 | - | 82.9/77.8 | 85.3 |
> | **Reasoning models** | | | | | | | | | | |
> | R1-VL-7B | 52.2 | 28.2 | 74.3 | - | - | - | - | **2395.0** | - | - |
> | Vision-R1-7B | 52.4 | - | 73.5 | - | - | - | - | - | - | - |
> | R1-OneVision-7B | 46.1 | 22.5 | 63.9 | 62.1 | 65.6 | - | - | - | - | - |
> | OpenVLThinker-7B | 48.0 | 25.0 | 71.5 | 67.8 | 70.8 | - | - | - | - | - |
> | MM-Eureka-7B | 50.5 | 28.3 | 71.5 | 65.5 | 68.3 | - | - | - | - | - |
> | ThinkLite-VL-7B | 50.2 | 27.6 | 72.7 | 69.2 | 71.0 | - | 81.4 | - | - | - |
> | VLAA-Thinker-7B | 49.9 | 26.9 | 68.8 | 67.9 | 68.6 | - | - | - | - | - |
> | NoisyRollout-7B* | 52.6 | 28.7 | 73.1 | 70.6 | 71.2 | 55.2 | 83.4 | 2376.2 | 58.4/48.1 | 67.0 |
> | Perception-R1-7B* | 50.1 | 28.4 | 73.3 | **72.6** | 68.6 | 55.1 | 83.0 | 2355.5 | **60.7**/50.6 | 66.5 |
> | VL-Rethinker-7B* | 52.6 | **30.6** | 74.2 | 68.1 | 70.1 | 55.0 | 82.4 | 2369.1 | 60.4/49.9 | 66.0 |
> | Base Model | 47.1 | 26.6 | 68.6 | 63.4 | 68.6 | 55.0 | **83.7** | 2332.7 | 58.4/45.5 | 66.0 |
> | GRPO | 51.1 | 27.9 | 71.0 | 68.2 | 68.9 | - | - | - | - | - |
> | **Ours(Geo3K)** | **53.3** | 28.6 | 74.6 | 71.7 | **71.5** | 54.6 | 83.6 | 2363.0 | 58.6/48.3 | 64.9 |
> | **Ours(ViRL39K)** | 52.1 | 29.7 | **75.2** | 71.6 | **71.5** | **56.6** | 83.2 | 2368.0 | 60.5/**51.1** | **69.1** |

---

> ### Author Response · Authors · 2025-11-23
>
> **W3:** The paper situates itself within the RLVR/R1 line but does not thoroughly compare with contemporaneous works such as VL-Rethinker or R1-VL that also incorporate self-reflective or vision-aligned objectives.
>
> **A**: We have enhanced our experimental analysis by including comparative evaluations with both VL-Rethinker and R1-VL.
>
> **Q1**: To strengthen the paper, the authors should (i) clarify the novelty boundary relative to that prior work, (ii) demonstrate robustness under open-source captioners and cheaper reward pipelines, and (iii) include a brief cross-domain experiment to confirm generality. Addressing these points would considerably improve both the clarity and contribution of the paper.
>
> **A**: Our model was trained on the ViRL39K dataset using an open-source and cost-effective approach: we employed Qwen3-VL-32B-Instruct to generate image captions in place of Gemini-2.5-Pro. Experimental results demonstrate the effectiveness and robustness of our proposed method.

---

### Official Review · Reviewer_nXHk · 2025-11-01

**Soundness:** 3
**Presentation:** 3
**Contribution:** 3
**Rating:** 4
**Confidence:** 2

**Summary:**

RLVR boosts MLLM reasoning but leaves perception errors unaddressed. The authors show via error analysis that after GRPO training, perception errors dominate failures. They propose a perception-first approach: the model generates a before reasoning, rewarded by semantic consistency with PGT captions (from Gemini) judged by an LLM. This perception reward is multiplicatively combined with accuracy reward (logical AND). Using modified GRPO on Qwen2.5-VL-7B, they achieve SOTA or near-SOTA on MathVerse (53.3%), MathVista (74.6%), We-Math, and HallusionBench (71.5%), confirming that fixing perception unlocks better multimodal reasoning.

**Strengths:**

The paper convincingly identifies perception as the dominant post-RLVR bottleneck through a rigorous 200-sample error audit and addresses it with an elegant, scalable solution: a caption-first protocol paired with a fine-grained perception reward derived from PGT captions and an LLM judge, multiplicatively fused with accuracy to enforce joint correctness. This principled design yields consistent, state-of-the-art, or near-SOTA gains across five diverse multimodal benchmarks, strongly validated by clean ablations and clear presentation.

**Weaknesses:**

- Evaluation is limited to math reasoning, leaving generalization to broader multimodal tasks untested. Training on only 2.1K samples raises scalability concerns, and ablation results are inconsistent (e.g., no gain on We-Math, drop on MathVision), undermining claims of universal perception improvement.

- Perception is only measured on HallusionBench. HR-Bench (high-resolution hallucination) and Vstar Bench (video+chart perception) are standard for visual reliability, yet are absent.
- The claim “perception bottleneck solved” remains unproven outside toy geometry diagrams.

**Questions:**

Please refer to the weaknesses.

---

> ### Author Response · Authors · 2025-11-23
>
> We sincerely thank you for your valuable comments, which have been instrumental in helping us improve the paper. Below is a summary of the revisions we have made. Please let us know if these address your concerns adequately.
>
> **W1**: Evaluation is limited to math reasoning, leaving generalization to broader multimodal tasks untested. Training on only 2.1K samples raises scalability concerns, and ablation results are inconsistent (e.g., no gain on We-Math, drop on MathVision), undermining claims of universal perception improvement.
>
> **A**: We have expanded our evaluation to include multiple benchmarks—such as BLINK, MMBench, MME, HR-Bench, and VstarBench—to comprehensively assess the performance of our method. Additionally, we trained our model on the ViRL39K dataset, which contains 39K samples. The results presented below demonstrate the effectiveness of our approach across these diverse benchmarks. Ablation studies are currently underway and will be supplemented in a future version due to time constraints.
>
> | **Model** | **MathVerse** | **MathVision** | **MathVista** | **WeMath** | **HallusionBench** | **BLINK** | **MMBench1.1(EN)** | **MME(sum)** | **HRBench(4K/8K)** | **VstarBench** |
> |:---:|:---:|:---:|:---:|:---:|:---:|:---:|:---:|:---:|:---:|:---:|
> | **Close-source models** | | | | | | | | | | |
> | GPT-4o | 50.8 | 30.4 | 63.8 | 69.0 | 71.4 | - | 82.1 | - | - | - |
> | Claude-3.5-Sonnet | 26.5 | 38.0 | 67.7 | - | 71.6 | - | 83.4 | - | - | - |
> | Kimi-k1.5 | - | 38.6 | 74.9 | - | - | - | - | - | - | - |
> | Gemini2.5-pro | 65.9 | 66.0 | 77.7 | - | 60.9 | 70.0 | 86.6 | - | - | - |
> | **Open-source models** | | | | | | | | | | |
> | LLaVA-OneVision-7B | 26.2 | - | 63.2 | - | 48.4 | 48.2 | 80.8 | - | - | - |
> | Mulberry-7B | - | - | 63.1 | - | - | - | - | 2396.0 | - | - |
> | InternVL2.5-8B | 39.5 | 19.7 | 64.4 | - | 67.3 | 54.8 | 83.2 | 2344.1 | - | - |
> | URSA-8B | 45.7 | 26.2 | 59.8 | - | - | - | - | - | - | - |
> | Kimi-VL-16B | 44.9 | 21.4 | 68.7 | - | 66.2 | 57.3 | 83.1 | - | - | - |
> | Qwen2.5-VL-72B-Instruct | - | 38.1 | 74.8 | - | 71.9 | - | 88.0 | 2448.0 | - | - |
> | InternVL2.5-78B | 51.7 | 32.2 | 72.3 | - | 72.9 | 63.8 | 88.5 | 2494.5 | - | - |
> | Qwen3-VL-32B-Instruct | 76.8 | 63.4 | 83.5 | - | 63.8 | 67.3 | 88.9 | - | 82.9/77.8 | 85.3 |
> | **Reasoning models** | | | | | | | | | | |
> | R1-VL-7B | 52.2 | 28.2 | 74.3 | - | - | - | - | **2395.0** | - | - |
> | Vision-R1-7B | 52.4 | - | 73.5 | - | - | - | - | - | - | - |
> | R1-OneVision-7B | 46.1 | 22.5 | 63.9 | 62.1 | 65.6 | - | - | - | - | - |
> | OpenVLThinker-7B | 48.0 | 25.0 | 71.5 | 67.8 | 70.8 | - | - | - | - | - |
> | MM-Eureka-7B | 50.5 | 28.3 | 71.5 | 65.5 | 68.3 | - | - | - | - | - |
> | ThinkLite-VL-7B | 50.2 | 27.6 | 72.7 | 69.2 | 71.0 | - | 81.4 | - | - | - |
> | VLAA-Thinker-7B | 49.9 | 26.9 | 68.8 | 67.9 | 68.6 | - | - | - | - | - |
> | NoisyRollout-7B* | 52.6 | 28.7 | 73.1 | 70.6 | 71.2 | 55.2 | 83.4 | 2376.2 | 58.4/48.1 | 67.0 |
> | Perception-R1-7B* | 50.1 | 28.4 | 73.3 | **72.6** | 68.6 | 55.1 | 83.0 | 2355.5 | **60.7**/50.6 | 66.5 |
> | VL-Rethinker-7B* | 52.6 | **30.6** | 74.2 | 68.1 | 70.1 | 55.0 | 82.4 | 2369.1 | 60.4/49.9 | 66.0 |
> | Base Model | 47.1 | 26.6 | 68.6 | 63.4 | 68.6 | 55.0 | **83.7** | 2332.7 | 58.4/45.5 | 66.0 |
> | GRPO | 51.1 | 27.9 | 71.0 | 68.2 | 68.9 | - | - | - | - | - |
> | **Ours(Geo3K)** | **53.3** | 28.6 | 74.6 | 71.7 | **71.5** | 54.6 | 83.6 | 2363.0 | 58.6/48.3 | 64.9 |
> | **Ours(ViRL39K)** | 52.1 | 29.7 | **75.2** | 71.6 | **71.5** | **56.6** | 83.2 | 2368.0 | 60.5/**51.1** | **69.1** |
>
> **W2**: Perception is only measured on HallusionBench. HR-Bench (high-resolution hallucination) and Vstar Bench (video+chart perception) are standard for visual reliability, yet are absent.
>
> **A**: We performed additional experiments and extended our evaluation to include HR-Bench and VstarBench.
>
> **W3**: The claim “perception bottleneck solved” remains unproven outside toy geometry diagrams.
>
> **A**: We trained model on the ViRL39K dataset, which encompasses a wide variety of data types, including non-geometric math, geometric math, diagrams, charts, documents, spatial relationships, and others. The model achieved strong results across multiple benchmarks.

---

### Official Review · Reviewer_9mSd · 2025-11-01

**Soundness:** 2
**Presentation:** 3
**Contribution:** 2
**Rating:** 4
**Confidence:** 3

**Summary:**

This paper addresses the challenge of enhancing multimodal reasoning capabilities in Multimodal Large Language Models (MLLMs) through perception reward modeling. Similar to Perception-R1, the authors recognize that existing Reinforcement Learning with Verifiable Rewards (RLVR) methods focusing solely on final answer correctness overlook the critical role of visual perception in multimodal reasoning. The paper proposes a perception reward modeling approach to explicitly guide MLLMs toward improving their visual understanding capabilities during reinforcement learning training.

**Strengths:**

1. Addresses fundamental perception bottleneck in RLVR for mm reasoning
2. Systematic reward modeling framework: Framework for incorporating visual understanding into the reinforcement learning objective, going beyond simple accuracy-based rewards.
3. Comprehensive experimental validation: Multiple benchmarks demonstrating improvements

**Weaknesses:**

1. Insufficient direct perception evidence : needs evaluation on dedicated perception benchmarks (BLINK, MMBench, MME)
2. Unclear distinction from Perception-R1[1] : both papers appear to address the same problem with similar approaches
Potential knowledge distillation conflation
3. Improvements may come from teacher model distillation rather than pure perception enhancement,  can you prove perception improvements rather than distillation effect?
3. Limited mechanistic analysis : needs deeper investigation of reward modeling dynamics. How were reward formulations chosen and validated?


[1] https://arxiv.org/abs/2506.07218

**Questions:**

See weakness.

---

> ### Author Response · Authors · 2025-11-23
>
> We sincerely appreciate your valuable feedback, which has been crucial in improving our manuscript. Below we provide point-by-point responses to your comments. Please do not hesitate to inform us if any further revisions are needed.
>
> **W1**: Insufficient direct perception evidence : needs evaluation on dedicated perception benchmarks (BLINK, MMBench, MME)
>
> **A**: We added evaluation using perceptual benchmarks such as BLINK, MMBench, and MME. The experimental results are shown in the table below.
>
> | **Model** | **MathVerse** | **MathVision** | **MathVista** | **WeMath** | **HallusionBench** | **BLINK** | **MMBench1.1(EN)** | **MME(sum)** | **HRBench(4K/8K)** | **VstarBench** |
> |:---:|:---:|:---:|:---:|:---:|:---:|:---:|:---:|:---:|:---:|:---:|
> | **Close-source models** | | | | | | | | | | |
> | GPT-4o | 50.8 | 30.4 | 63.8 | 69.0 | 71.4 | - | 82.1 | - | - | - |
> | Claude-3.5-Sonnet | 26.5 | 38.0 | 67.7 | - | 71.6 | - | 83.4 | - | - | - |
> | Kimi-k1.5 | - | 38.6 | 74.9 | - | - | - | - | - | - | - |
> | Gemini2.5-pro | 65.9 | 66.0 | 77.7 | - | 60.9 | 70.0 | 86.6 | - | - | - |
> | **Open-source models** | | | | | | | | | | |
> | LLaVA-OneVision-7B | 26.2 | - | 63.2 | - | 48.4 | 48.2 | 80.8 | - | - | - |
> | Mulberry-7B | - | - | 63.1 | - | - | - | - | 2396.0 | - | - |
> | InternVL2.5-8B | 39.5 | 19.7 | 64.4 | - | 67.3 | 54.8 | 83.2 | 2344.1 | - | - |
> | URSA-8B | 45.7 | 26.2 | 59.8 | - | - | - | - | - | - | - |
> | Kimi-VL-16B | 44.9 | 21.4 | 68.7 | - | 66.2 | 57.3 | 83.1 | - | - | - |
> | Qwen2.5-VL-72B-Instruct | - | 38.1 | 74.8 | - | 71.9 | - | 88.0 | 2448.0 | - | - |
> | InternVL2.5-78B | 51.7 | 32.2 | 72.3 | - | 72.9 | 63.8 | 88.5 | 2494.5 | - | - |
> | Qwen3-VL-32B-Instruct | 76.8 | 63.4 | 83.5 | - | 63.8 | 67.3 | 88.9 | - | 82.9/77.8 | 85.3 |
> | **Reasoning models** | | | | | | | | | | |
> | R1-VL-7B | 52.2 | 28.2 | 74.3 | - | - | - | - | **2395.0** | - | - |
> | Vision-R1-7B | 52.4 | - | 73.5 | - | - | - | - | - | - | - |
> | R1-OneVision-7B | 46.1 | 22.5 | 63.9 | 62.1 | 65.6 | - | - | - | - | - |
> | OpenVLThinker-7B | 48.0 | 25.0 | 71.5 | 67.8 | 70.8 | - | - | - | - | - |
> | MM-Eureka-7B | 50.5 | 28.3 | 71.5 | 65.5 | 68.3 | - | - | - | - | - |
> | ThinkLite-VL-7B | 50.2 | 27.6 | 72.7 | 69.2 | 71.0 | - | 81.4 | - | - | - |
> | VLAA-Thinker-7B | 49.9 | 26.9 | 68.8 | 67.9 | 68.6 | - | - | - | - | - |
> | NoisyRollout-7B* | 52.6 | 28.7 | 73.1 | 70.6 | 71.2 | 55.2 | 83.4 | 2376.2 | 58.4/48.1 | 67.0 |
> | Perception-R1-7B* | 50.1 | 28.4 | 73.3 | **72.6** | 68.6 | 55.1 | 83.0 | 2355.5 | **60.7**/50.6 | 66.5 |
> | VL-Rethinker-7B* | 52.6 | **30.6** | 74.2 | 68.1 | 70.1 | 55.0 | 82.4 | 2369.1 | 60.4/49.9 | 66.0 |
> | Base Model | 47.1 | 26.6 | 68.6 | 63.4 | 68.6 | 55.0 | **83.7** | 2332.7 | 58.4/45.5 | 66.0 |
> | GRPO | 51.1 | 27.9 | 71.0 | 68.2 | 68.9 | - | - | - | - | - |
> | **Ours(Geo3K)** | **53.3** | 28.6 | 74.6 | 71.7 | **71.5** | 54.6 | 83.6 | 2363.0 | 58.6/48.3 | 64.9 |
> | **Ours(ViRL39K)** | 52.1 | 29.7 | **75.2** | 71.6 | **71.5** | **56.6** | 83.2 | 2368.0 | 60.5/**51.1** | **69.1** |
>
> **W2**: Unclear distinction from Perception-R1[1] : both papers appear to address the same problem with similar approaches Potential knowledge distillation conflation
> [1] https://arxiv.org/abs/2506.07218
>
> **A**: Our work shares some similarities with [1], but differs in the following key aspects:
>
> 1. Unlike [1], which distills both the visual and reasoning capabilities of Gemini-2.5-Pro, we only leverage its visual ability to generate pseudo ground-truth visual descriptions of images.
> 2. Our approach first encourages the model to produce accurate visual perception (i.e., "caption → reasoning → answer"), and then enhances learning through a multiplicative coupling mechanism that combines visual perception rewards with answer rewards.

---

> ### Author Response · Authors · 2025-11-23
>
> **W3**: Improvements may come from teacher model distillation rather than pure perception enhancement, can you prove perception improvements rather than distillation effect?
>
> **A**: We distilled Gemini-2.5-Pro via supervised fine-tuning (SFT). Evaluation results on perception benchmarks show that while distillation yielded some improvements, our RL-trained model achieved even stronger perceptual capabilities.
>
> |**Model**|**MathVerse**|**MathVision**|**MathVista**|**WeMath**|**HallusionBench**|**BLINK**|**MMBench1.1(EN)**|**MME(sum)**|**HRBench(4K/8K)**|**VstarBench**|
> |:---:|:---:|:---:|:---:|:---:|:---:|:---:|:---:|:---:|:---:|:---:|
> |Base Model|47.1|26.6|68.6|63.4|68.6|55.0|**83.7**|2332.7|58.4/45.5|66.0|
> |SFT|36.2|21.1|65.9|54.5|68.8|54.0|81.6|2356.4|59.4/49.1|63.4|
> |**Ours(Geo3K)**|**53.3**|28.6|74.6|**71.7**|**71.5**|54.6|83.6|2363.0|58.6/48.3|64.9|
> |**Ours(ViRL39K)**|52.1|**29.7**|**75.2**|71.6|**71.5**|**56.6**|83.2|**2368.0**|**60.5/51.1**|**69.1**|
>
> **W4**: Limited mechanistic analysis : needs deeper investigation of reward modeling dynamics. How were reward formulations chosen and validated?
>
> **A**: We observed that directly rewarding image captions generated by Gemini-2.5-Pro and the policy model led to reward distributions overly concentrated in the mid-range. We hypothesize that the Judge LLM, possibly to avoid penalties, tends to assign scores around the middle of the scale. This behavior could undermine the effectiveness of RL training. Therefore, we subsequently experimented with splitting the captions, rewarding the segments separately, and then averaging the rewards. This approach proved highly effective. Due to time constraints, related experimental results are still under investigation.

---

### Author Response · Authors · 2025-12-02
**Summary of Paper Revision**

We sincerely thank all reviewers for their valuable and constructive comments. We address questions and concerns in detail below. Based to these comments, we have also revised the manuscript accordingly, with all changes marked in blue. The main revisions are summarized as follows:

**Additional experiments (Appendix C, to 9mSd, nXHk, Yw6V, and ay8z):**

We expanded the experimental section by training the QWen2.5-VL-7B-Instruct model on the larger-scale ViRL39K dataset using our proposed method. We also included comparisons with VL-Rethiner, R1-VL, Gemini2.5-pro, and Qwen3-VL-32B-Instruct, and evaluated performance across multiple perceptual benchmarks—such as BLINK, MMbench, MME, HR-bench, and VstarBench. These results further validate the effectiveness and generalization ability of our approach.

**Clarification of contributions and related work (Appendix D, to 9mSd and Yw6V):**

We have further clarified the distinctions between our method and contemporaneous works (e.g., perception-R1). Our key contribution is a framework that encourages the model to generate accurate visual perceptions through a "description → reasoning → answer" process. We then employ a multiplicative mechanism to integrate visual perception rewards with correctness rewards, thereby enhancing learning efficacy.

---

### Meta-Review · Area_Chair_rwqg · 2026-01-06

**Summary:**

The main concerns from the reviewers are following:

**W1**. The main idea overlaps with existing work [1].  (Reviewer **9mSd, Yw6V**)

**W2**. The experiments lack experiments for more diverse benchmarks and baselines. (Reviewer **9mSd,  nXHk, Yw6V, ay8z**)

[1] Perception-R1: Advancing Multimodal Reasoning Capabilities of MLLMs via Visual Perception Reward (https://arxiv.org/abs/2506.07218)

**Reviewer Concerns:**

**Concerns still remaining**:

- **W1**: In the rebuttal, the authors have discussed the differences between the two works. While in my view, the differences lie in details of implementation other than major ideas and insights.

- **W2**: The authors have provided additional experimental results on more benchmarks (in special the ones beyond math reasoning) and baselines. Even though the results show performance gains, the results are incomplete with a number of empty entries, including some key results, such as the missing ones of the GRPO baseline.

**Reviewer Scores:**

Due to the remaining concerns as discussed above, in my view, all the reviewers would not change their scores.

---

### Decision · Program_Chairs · 2026-01-26

Reject